# Dendritic Cell Vaccines: A Shift from Conventional Approach to New Generations

**DOI:** 10.3390/cells12172147

**Published:** 2023-08-25

**Authors:** Kyu-Won Lee, Judy Wai Ping Yam, Xiaowen Mao

**Affiliations:** 1Department of Pathology, School of Clinical Medicine, Li Ka Shing Faculty of Medicine, The University of Hong Kong, Hong Kong; kwlee48@pathology.hku.hk (K.-W.L.); judyyam@pathology.hku.hk (J.W.P.Y.); 2State Key Laboratory of Liver Research, The University of Hong Kong, Hong Kong; 3State Key Laboratory of Quality Research in Chinese Medicine, Institute of Chinese Medical Sciences, University of Macau, Macao

**Keywords:** dendritic cells, cancer immunotherapy, DC vaccines

## Abstract

In the emerging era of cancer immunotherapy, immune checkpoint blockades (ICBs) and adoptive cell transfer therapies (ACTs) have gained significant attention. However, their therapeutic efficacies are limited due to the presence of cold type tumors, immunosuppressive tumor microenvironment, and immune-related side effects. On the other hand, dendritic cell (DC)-based vaccines have been suggested as a new cancer immunotherapy regimen that can address the limitations encountered by ICBs and ACTs. Despite the success of the first generation of DC-based vaccines, represented by the first FDA-approved DC-based therapeutic cancer vaccine Provenge, several challenges remain unsolved. Therefore, new DC vaccine strategies have been actively investigated. This review addresses the limitations of the currently most adopted classical DC vaccine and evaluates new generations of DC vaccines in detail, including biomaterial-based, immunogenic cell death-inducing, mRNA-pulsed, DC small extracellular vesicle (sEV)-based, and tumor sEV-based DC vaccines. These innovative DC vaccines are envisioned to provide a significant breakthrough in cancer immunotherapy landscape and are expected to be supported by further preclinical and clinical studies.

## 1. Introduction

Immunotherapy has rapidly advanced for clinical cancer treatment, highlighting the importance of tumor–immune interactions in tumor regulation [1,2]. Despite recent significant progress in immune checkpoint blockades (ICBs) and adoptive cell therapies (ACTs) across a broad range of solid tumors, limited therapeutic efficacies and severe immune-related adverse events have rendered them from wide applications [3,4]. The highly immunosuppressive tumor microenvironment (TME), in which the numbers and functions of immunostimulatory cells such as effector T cells (T_EFF_) and antigen-presenting cells (APCs) are downregulated, remains one of the key determinant factors that subverts the therapeutic efficacies of existing cancer immunotherapies [5,6]. Although the cancer immunotherapy landscape is currently dominated by ICBs and ACTs, cancer vaccines that are supported by clear rationale and encouraging preclinical data for further development also appear promising [7,8]. 

Cancer vaccines carry distinct benefits that address the limitations of currently popular ICBs and ACTs. They can target an additional broader set of intracellular antigens, unlike ACTs that target distinguished tumor-specific surface antigens and require optimal target selection to circumvent antigen escape and “on-target off-tumor” toxicity [9,10]. They can also prime naïve tumor-reactive CD8^+^ T cells unlike ICBs, which are only responsive in patients with pre-existing antitumor immunity [11,12]. Cancer vaccines can also be implemented as a part of multimodal immunotherapeutic regimens. 

Therapeutic cancer vaccines are designed to stimulate the patient’s antigen-specific adaptive immune response to eliminate cancer cells and generate durable patient responses [13]. Their purpose to eradicate cancer cells via antigen-specific cellular immunity is what makes them different from traditional preventative vaccines [14,15]. Cancer vaccine activity is mostly dependent on activated antigen-specific CD8^+^ T cells that differentiate into CD8^+^ T_EFF_ or cytotoxic T lymphocytes (CTLs) to reject cancer cells [15]. Therefore, vaccine-elicited CD8^+^ T cells should ideally be of high T cell avidity or antigen-triggering sensitivity to recognize immunogenic peptide-major histocompatibility complex class I (pMHC I) complexes and effectively eliminate cancer cells [16,17]. In addition, vaccination should generate long-lived memory CD8^+^ T cells to prevent tumor relapse [18]. To achieve the ideal vaccine-elicited immune responses, the protective role of immune system components, including antigen presentation by appropriate APCs, induction of CD4^+^ T_EFF_ or T helper (Th) cells, and suppression of immunosuppressive regulatory T cells (Treg) and TME, are also required [18,19,20].

Dendritic cells (DCs) are the most potent group of specialized APCs with key roles in initiation and regulation of innate and adaptive immune responses [21,22]. DCs comprise heterogenous populations, which can be broadly classified into conventional DCs (cDCs), consisting of two subsets (cDC1, cDC2), plasmacytoid DCs (pDCs) and monocyte-derived DCs (MoDCs), based on their ontogeny [23,24,25]. Recently, additional subsets and states of DCs such as DC3 are being revealed with advances in high-throughput single-cell analysis, which would facilitate a more accurate understanding of the functions and development of DCs [26,27,28,29,30,31,32,33,34]. As DCs initiate and modulate antigen-specific immunity and tolerance, the exploitation of their antigen-presenting capacities and heterogeneity provides great potential for improving antitumor immune response elicited by therapeutic vaccines [7,18,35]. 

DCs patrol their environment until they become activated upon encountering foreign pathogens or altered cells such as cancer cells as illustrated in Figure 1 [36]. Exogeneous danger signals, such as pathogen-associated molecular patterns (PAMPs) and damage-associated molecular patterns (DAMPs), activate and determine DC functions via pattern recognition receptors (PRRs), including Toll-like receptors (TLRs) [37,38]. DCs then migrate to tumor-draining lymph nodes (TDLNs) and perform cross-presentation, which is a process where the acquired exogenous antigens are presented on MHC I molecules [39]. Activated, mature, antigen-loaded DCs cross-prime naïve T cells into antigen-specific T_EFF_ cells, including CD4^+^ Th cells and CD8^+^ CTLs via T cell receptors (TCRs), and regulate immunogenic responses depending on the DC-derived cytokine environment [40,41,42,43]. Cross-presentation is essential to activate and cross-prime CTLs for defense against tumors [44]. However, the challenge to enhance cross-presentation abilities of DCs for effective cross-priming of CTLs often remains unmet in cancer immunotherapy [39,45]

## 2. Classical DC Vaccine

The two conventionally adapted methods for preparing DC-based vaccination are ex vivo differentiation of DCs from CD14^+^ monocyte precursors or CD34^+^ hematopoietic stem and progenitor cells (HSPCs) and direct targeting of antigens to DCs in vivo [20,46]. 

Reinfusion of ex vivo manipulated DCs is the most explored preparation method of DC-based vaccines, which is used in approximately 97% of clinical trials [47]. In this approach, CD14^+^ monocytes or CD34^+^ HSPCs are collected from patients by leukapheresis, differentiated into immature DCs in the presence of granulocyte macrophage-colony stimulating factor (GM-CSF) and interleukin-4 (IL-4), and simultaneously pulsed with tumor-associated antigens (TAAs) or tumor cell lysates while being stimulated in a maturation cocktail (Figure 2A) [46,48,49,50]. In most clinical studies, CD14^+^ monocytes are preferentially used to be differentiated into MoDCs because it is relatively easier to collect enough CD14^+^ monocytes as they represent 10% of peripheral blood mononuclear cells (PBMCs) [47]. Although CD34^+^-derived heterogenous APC populations were shown to stimulate CTLs more significantly than MoDCs (NCT00700167, NCT01456104), the limited numbers of CD34^+^ HSPCs (represent 0.1% of PBMCs) that can be isolated from apheresis products hinder them from clinical applications [51,52,53]. Due to the absence of standardized preparation of ex vivo manipulated DC vaccines, different studies follow various methods for DC sourcing, maturation, antigen loading, and administration [47]. So far, only sipuleucel-T (Provenge), the first FDA-approved therapeutic cancer vaccine, has demonstrated a satisfactory efficacy in phase III trials as an autologous ex vivo DC vaccine for metastatic castration-resistant prostate cancer (NCT00065442) [54,55,56]. The combination of Provenge with ICBs and a homeostatic cytokine IL-7 that enhances T and B cell development and proliferation has demonstrated encouraging clinical efficacies (Phase I, NCT01832870; Phase II, NCT01804465, NCT01881867) [57,58,59]. 

Another strategy that aims to directly target antigens to endogenous DCs in vivo has been explored to overcome the limitations of ex vivo manipulated DC vaccines [60]. This strategy involves antigen coupling to monoclonal antibodies (mAb) that are specific to DC surface molecules, including Clec9A, CD40, or DEC-205, to directly pulse DCs with antigens in vivo [61,62,63,64,65]. Despite effective T_EFF_-mediated antitumor responses and humoral immunity demonstrated in preclinical and clinical studies, clinical implementations have been discouraging [66]. Some of the drawbacks of this approach may be the requirement to ensure expressions of targeted receptor in the selected DC subpopulation and to co-administer DC maturation agents to prevent antigen tolerance in steady state [46,64,67].

## 3. Limitations of Classical DC Vaccine

Despite the early success of Provenge and generally accepted safety of conventional DC-based cancer vaccines, their clinical implications have generally been unsuccessful, with only 5–15% of patients benefiting from an objective immune response [68]. The limited efficacy of cancer vaccines may be mainly due to the presence of multiple immunosuppressive factors in the TME that act as immune rheostats or immunostats in short. In other words, the immunosuppressive factors modulate the antitumor T cell responses and act as a common rate-limiting step in clinical studies that altogether limit the clinical efficacies of cancer vaccines [69]. New generations of DC vaccines must therefore overcome multiple immunosuppressive mechanisms in the TME to improve T_EFF_-mediated antitumor responses. 

Solid tumors produce soluble immunosuppressive mediators, such as indoleamine 2,3-dioxygenase (IDO), transforming growth factor beta (TGF-β), and vascular endothelial growth factor (VEGF), which not only suppress Th1 cells and CTL activity but also impair DC functions [70,71,72]. In addition, the TME attracts immunomodulatory populations such as Tregs and myeloid derived suppressor cells (MDSCs) and suppresses TAA expression to achieve immune evasion [73,74,75]. Tumor cells may also evade immune recognition by the overexpression of immune checkpoints such as cytotoxic T-lymphocyte antigen-4 (CTLA-4) and programmed cell death protein 1 (PD-1) [76]. Therefore, combination therapies incorporating ICBs and DC-based vaccines are being actively studied as potential improved therapeutic regimen (NCT01067287, NCT03035331, NCT04203901). Altogether, it is important to note that the effectiveness of DC vaccines may be altered post-administration of in vitro-modulated DCs to an in vivo immunosuppressive environment [21]. 

Despite MoDCs being the most adopted subset in clinical trials for DC vaccines, it remains unclear which subset is the most effective. Ex vivo generated MoDCs are suggested to be functionally dissimilar to the steady-state DC subsets present in the body, revealed by the disparities between their immune system-related transcripts at both transcriptional and phenotypical levels [20,77]. Moreover, the immunosuppressive TME limits the cross-presentation and T cell stimulation abilities of tumor-associated MoDCs [78]. In fact, studies suggest that endogenous DCs are required for T cell priming, as the capacity of ex vivo manipulated MoDCs as effective APCs in vivo is obscure [79]. In addition, MoDCs in the TME have limited capacity in migration to TDLNs [80,81]. The optimization of effective DC migration is thus another obstacle in the conventional DC vaccine regimen. To circumvent the complex migration cascade required for efficient homing of DCs to the TDLNs, in situ targeting of DCs has been proposed as an attractive alternative [82]. 

Therefore, a new generation of DC-based vaccines that incorporates heterogenous subsets of DCs to resemble the in vivo environment more closely may be beneficial [83]. For instance, both cDCs and pDCs have the potential to directly activate T cells in certain environmental cues [83]. Therefore, vaccines that can enhance the function of all DC subsets and cells that crosstalk with them may maximize tumor-specific T cell priming efficacy and vaccine longevity. In addition, ex vivo manipulation of MoDCs also requires extensive in vitro culture, which often disturbs their functionalities [68]. On top of this, despite the good safety profile of MoDC-based vaccines, their production according to good manufacturing procedure is highly expensive and laborious with inconsistent success [47]. The ultimate goal is to design new generations of immunotherapies that sustainably increase the magnitude of DC cross-presentation presentation and T_EFF_-derived antitumor immunity. Figure 2B illustrates the recent advances in DC-based vaccine regimen, which are discussed in detail below.

## 4. Biomaterial-Based DC Vaccines

The limited efficacies of conventional ex vivo differentiated DC vaccines were partially due to their inefficient migration to TDLNs and cross-presentation abilities when compared to endogenous DCs [51,52,81]. To overcome such limitations, several biomaterial-based strategies that directly target antigens to endogenous DCs have been developed [84]. These new strategies consist of implantable or injectable biomaterial-based scaffolds that allow in situ trafficking and the modulation of endogenous DCs [46]. The biomaterials are established to provide a spatiotemporally controlled delivery of chemoattractants, antigens, and adjuvants to recruit and activate desired endogenous DC populations [85]. The biomaterial-based scaffolds also regulate the kinetics of antigen exposure, which is critical in developing antigen-specific T cell response [86,87,88]. Moreover, unlike the traditional bolus vaccination strategy that delivers vaccine components in suspension over a short period of time, biomaterial-based scaffolds offer a new biomimetic immunogenic microenvironment in the body where DCs are activated and sustainably provided with antigens and stimulatory molecules over a period of at least two weeks [89,90].

Several implantable three-dimensional biomaterial-based scaffolds are based on a porous structure composed of poly(lactide-coglycolide) (PLG) [89]. PLG is FDA-approved for clinical use, prone to surface modification to enhance its unique scaffold characteristics, biocompatible, and biodegradable [91]. It has been shown that PLG scaffolds can encapsulate DC chemoattractants such as GM-CSF, and gradually release them over 15 days post-implantation [89,92]. In addition, PLG matrices are able to immobilize CpG-rich oligodeoxynucleotides (CpG-ODN), which are agonists of TLR9, as danger signals to activate DCs [9,93]. The spatiotemporal delivery of GM-CSF and CpG-ODN formulation via PLG matrices have successfully attracted and activated DCs in situ and induced prophylactic immunity against inoculations of murine B16-F10 melanoma cells [89]. 

In a subsequent study, a PLG scaffold that contained GM-CSF, CpG-ODN, and melanoma tumor lysates were assayed in B16-F10 murine melanoma models. Of note, the scaffold successfully recruited and activated heterogenous DC subsets, both pDCs and CD8^+^ cDCs [94]. Moreover, simultaneous trafficking and activation of pDCs and cDCs led to local accumulation of CTLs and superior antitumor responses than targeting a single DC subset [83]. Conventional ex vivo-developed cancer vaccines are unable to fully recapitulate the broad DC responses in vivo. Therefore, in situ generation of a heterogenous DC subsets by implantable immunostimulatory PLG matrices to activate robust CTL responses against established tumors is remarkable [94]. In addition, two-time vaccination with the PLG matrices led to a complete regression of established melanoma in ~47% of mice in preclinical studies, supporting the therapeutic efficacies of PLG scaffold vaccines [94]. The PLG matrices also increased the local production of IL-12, a Th1-promoting cytokine, which may be attributed to the CD8^+^ cDCs recruited that are adept at IL-12 production and CTL induction [95,96,97]. Vaccine efficacy highly correlated to the quantities of recruited CD8^+^ cDCs and pDCs, together with local GM-CSF and IL-12 concentrations [98]. Of note, the accumulation of CD8^+^ cDCs at the vaccine site is a marked feature of this strategy as CD8^+^ cDCs typically localize at secondary lymphoid structures [99]. It is suggested that the recruited pDCs supported the activation of CD8^+^ cDCs and priming of Th1 cells and CTLs, thereby inducing prophylactic immunity against tumor. In contrast to other vaccines that only incorporated GM-CSF as an adjuvant and showed a significant Treg increase at the vaccine site, the PLG matrix also incorporating CpG-ODN counteracted the immunosuppressive mechanisms as CD8^+^ CTLs outnumbered Tregs [100]. This acellular PLG scaffold in concert with GM-CSF, CpG-ODN, and TAAs may also be adapted for other types of solid cancers. A human version of this vaccine, named WDVAX, is currently in a phase I clinical trial for stage IV melanoma (NCT01753089) [90]. Furthermore, a combination of this PLG matrix with anti-CTLA-4 mAbs augmented vaccine-induced CTL activity and overall survival and led to tumor regression in B16-F10 melanoma mouse models [101]. This combination approach also obviated the need for multiple PLG vaccination, which may be attributed to the synergistic activation of multiple DC subsets and tumor-infiltrating CD8^+^ T cells [101]. 

The biomaterial-based vaccination system has expanded to injectable biomaterials such as hydrogels [46,102,103,104]. Alginate hydrogels are often obtained by cryogelatination, a technique which augments mechanical stability, pore connectivity, and shape memory to the injected hydrogels [105,106,107,108]. Injectable alginate gels are biocompatible and elicit a mild host response in vivo while providing sustained release of molecules [109,110,111]. In addition, these gels do not require surgical implantation, unlike PLG scaffold vaccines, to provide a niche for immunomodulation [112]. One of the challenges of ex vivo differentiated DC vaccine is the short half-life of pMHC on short-lived DC [113]. The alginate matrix may act as a local depot for antigens or pMHC to attract local DCs and sustain T_EFF_ cell stimulation. Notably, the total number of local DCs recruited into the alginate gel was sustained [114]. Moreover, similar to the PLG scaffold that delivered GM-CSF and CpG-ODN, alginate hydrogel loaded with irradiated B16-F10 melanoma cells, GM-CSF, and CpG-ODN also elicited local infiltrations of cDCs and pDCs into the gel in a spatiotemporal manner in syngeneic mice [89,115]. The accumulation of a larger spectrum of DC subsets in situ elicited by these biomaterial scaffolds confer sustained DC functions and numbers, in contrast to conventional adoptive transfer ex vivo vaccines [79,93,116]. 

Combinatory strategies of loading other immunomodulatory agents in hydrogel-based platforms such as metabolism inhibitors or checkpoint blockades were found to augment antitumor immune response [117,118,119]. Metabolism inhibitors are envisioned as alternatives to more toxic cytokines that can counteract the immunosuppressive T cell activities. For instance, epacadostat, a small molecule inhibitor of IDO that suppresses T_EFF_, activated CD8^+^ T cells and suppressed the proliferation of Tregs [72,120,121]. Spatiotemporal controlled delivery of metabolism inhibitors via biomaterial scaffolds facilitated in improving poor pharmacokinetics and metabolic interventions [117,118,122]. Peritumoral injection of a hydrogel encapsulating GM-CSF and epacadostat in mice with subcutaneous (s.c.) 4T1 breast cancer cells recruited greater intratumoral (i.t.) cDCs with a 12-fold increase and upregulated expressions of a DC maturation marker CD86 compared to those injected with blank gels [123]. Moreover, the hydrogel successfully regulated the TME, as increased CTLs/Treg ratio and upregulated expression of T cell activation and proliferation marker CD69 on tumor-infiltrating CD8^+^ T cells was observed [123]. However, epacadostat does not completely inhibit the production of kynurenine (Kyn), a tryptophan catabolite by IDO, which promotes Treg and tumor-associated macrophage differentiation. Thus, epacadostat has failed to improve survival in Phase III clinical trials (NCT02752074) [124,125,126,127]. To address this limitation, one study reported a synergistic effect of alginate gels and a localized chemo-immunometabolic therapy in augmenting T cell activity and systemic anti-tumor immunity in poorly immunogenic 4T1 breast cancer and B16-F10 melanoma mouse models [128]. Enzyme-mediated Kyn elimination and chemotherapeutics were achieved by integrating kynureninase (KYNase) and a chemotherapeutic drug doxorubin (DOX) in alginate gels loaded with GM-CSF and CpG-ODN [128,129,130]. Previous studies have demonstrated increased in situ DC recruitment and systemic tumor-specific CD8^+^ T cells upon injection of alginate gels loaded with GM-CSF, CpG-ODN, and DOX conjugate [131]. The combinatory approach with chemo-immunometabolic therapy via alginate-based DC vaccine extended the systemic antitumor response, as numbers of tumor-specific CD8^+^ T cells increased in TDLNs, which could also induce abscopal effect in untreated tumors [128].

The spatiotemporal control of tumor antigens and different immunoadjuvants by the biomimetic biomaterial-based scaffolds successfully recruited endogenous DCs in situ and produced a more robust antitumor response. The scaffolds may incorporate different immunostimulatory molecules, metabolism inhibitors, and/or checkpoint blockades to maximize DC recruitment and activation, along with tumor-specific T cell responses, as well as modulating other immunosuppressive cells to modulate the TME. Although some scaffolds require surgical implantation and multiple doses, the recent development of injectable 3D structures and combination therapies alleviates such limitations and bypasses multiple immunizations [132,133]. Further studies are expected to provide more promising results of clinical implications of biomaterial-based DC vaccines.

## 5. Combinatory Immunogenic Cell Death-Inducing DC Vaccines

Conventional DC vaccines encounter unresponsiveness in cold tumors in which DC recruitment is defective, and thus T cell priming, activation, and infiltration are impaired [134,135,136]. Recently, increasing numbers of studies have shown that the induction of immunogenic cell death (ICD) is promising in converting cold tumors into hot tumors, thereby improving the immunogenic potential of DC vaccines in cold tumors [137,138,139]. ICD is a form of stress-induced regulated cell death response via different organellar and cellular stressors [140,141,142]. ICD is induced in cancer therapy by most common clinical treatments such as chemotherapy and radiotherapy, or emerging therapies such as photodynamic therapy (PDT) and photothermal therapy (PTT) [143,144,145,146,147]. ICD-inducing therapy is an attractive antitumor strategy for various solid tumors in view of its consequent antigenicity and adjuvanticity [144,148,149,150]. The antigenicity of ICD is determined by the level of tumor antigens from dying tumor cells [151,152,153]. The induction of ICD stimulates tumor cells to secrete, release, or surface-expose DAMPs and cytokines as adjuvants or danger signals [154,155,156]. Once emitted by dying cells, DAMPs recruit and activate innate and adaptive immune cells to elicit a specific antitumor immune response [155,157,158,159]. Antigenicity and adjuvanticity of anticancer treatment-induced ICD are critical for the development of long-term immunological memory against residual cancer cells and metastatic cells [160,161,162].

Canonical immunostimulatory DAMPs include, but are not limited to, calreticulin (CRT), heat shock proteins 70 (HSP70), high-mobility group box 1 (HMGB1), and ATP, and are associated with immunogenicity induced by DCs [140,161]. CRT confers immunogenicity by promoting phagocytosis of dying cells by DCs as an ‘eat-me’ signal [163,164]. Exogenous ATP, on the other hand, constitutes a ‘find-me’ signal that promotes DC recruitment and migration [165,166]. HMGB1 promotes activation, maturation, and cross-presentation abilities of DCs, while HSP70 increases the uptake of tumor cells and stimulates the migration and maturation of DCs [167,168,169,170]. Therefore, ICD activates tumor-specific immunity by tipping the cancer-immunity balance towards antitumor immunity while eradicating cancer cells and generating antigens in situ, which is in line with the objectives of cancer vaccines [69,139,171].

Radiotherapy not only induces ICD and abolishes highly proliferative cancer cells, but also promotes an abscopal effect, which is a rare phenomenon of tumor regression outside the field of local therapy [135,172,173]. However, tumor-specific immune response and abscopal effect induced by standalone radiotherapy are limited in immunosuppressive TME [137,174]. Moreover, radiotherapy can further exacerbate negative regulatory mechanisms by insufficient T cell priming and enhanced levels of Tregs and MDSCs [175,176]. To overcome the complex interaction between TME and radiotherapy, a combination of radiotherapy and immunotherapy is envisioned to generate an in situ cancer vaccine and enhance antitumor responses and abscopal effects [177]. Local administrations of immunostimulatory agents, along with radiotherapy, are increasingly being investigated to activate immature tolerogenic DCs into immunogenic DCs in established tumors. Many preclinical studies provided evidence that TLR agonists targeting TLR3, TLR9, and TLR4 together with local radiotherapy led to enhanced antitumor immunity [178]. The TLR3 agonist poly-ICLC was utilized to activate endogenous DCs together with Fms-like tyrosine kinase 3 ligand (Flt3L) as an adjuvant along with local radiotherapy as a treatment strategy for indolent non-Hodgkin’s lymphoma (iNHL) [179]. Such a combination functioned as an in situ vaccine, inducing greater i.t. infiltration of CD8^+^ T cells, DC activation, and cross-presentation [179]. Similarly, i.t. administration of Flt3L, local radiotherapy, and TLR3 stimulation followed by surgical resection demonstrated a potential multimodal therapy that can generate tumor-specific CTLs, achieve improved systemic tumor remission from 40% to 80%, and delay metastases in highly metastatic tumors [180]. Of note, both regimens observed increased expressions of PD-1 on CD8^+^ T cells, and thus synergized with PD-1/PD-L1 blockade to improve systemic tumor control and prolong survival [179,180]. 

These preclinical studies prompted a phase I/II trial of i.t. injection of Flt3L and poly-ICLC combined with radiotherapy in lymphoma patients (NCT01976585), which consistently demonstrated increased levels of PD-1^+^ CD8^+^ T cells and clinical efficacies. Furthermore, to confirm that ICB and radiotherapy synergistically facilitate antitumor immune response, a follow-up phase I/II clinical trial (NCT03789097) was performed. This demonstrated tolerability and early signs of efficacies in patients with iNHL, metastatic breast cancer, and head and neck squamous cell carcinoma [181]. Several clinical studies have also demonstrated that the combination of TLR9 agonist CpG-ODN and radiotherapy in various cancer models achieved systemic antitumor immunity and patient responses (phase I/II, NCT00185965; phase II, NCT00880581) [182,183]. These studies altogether support the combination of radiotherapy and immunotherapy in inducing synergistic effects that restrain not only local tumor growth but also distant metastases. The expansion of radiotherapy combined with immunotherapy may yield superior antitumor immunity compared to combination immunotherapies, which may result in increased toxicity [179]. While the combination of radiotherapy with other ICBs, such as CTLA-4, has also been robustly investigated (phase I/II, NCT00323882, NCT02221739; phase III, NCT00861614), further preclinical and clinical studies will demonstrate the promise of various combination strategies to build a robust in situ radiotherapy-based DC vaccine [184,185,186,187].

PDT-induced therapeutic ICD strategies have also been shown to be effective in various types of cancer and were approved by FDA [188]. PDT involves the use of a photosensitizer which is excited by a light of specific wavelength corresponding to the absorption spectrum of the photosensitizer in the presence of oxygen [151,189]. This results in the generation of singlet oxygen and other cytotoxic oxidants that trigger ICD in cancer cells [190]. Photosense (PS), a clinically approved photosensitizer, was found to successfully trigger ICD in several cancer cell types, including GL261 glioma cells [191]. ICD induced by PS-mediated PDT (PS-PDT) emitted CRT, HMGB1, and ATP, which successfully matured and activated DCs in vitro that produced IL-6 for T cell priming [191,192]. To demonstrate the efficacy of PS-PDT as a vaccine regimen, DCs were loaded ex vivo with GL261 glioma cells undergoing ICD post-PS-PDT (DC-GL261_PS-PDT) [160]. DC-GL261_PS-PDT not only protected mice against orthotopic challenge with GL261 cells prophylactically but also protected them therapeutically [160]. Moreover, the vaccinated mice showed significantly lower tumor mass, later onset of symptoms, and a significantly increased survival compared to control mice which were vaccinated with either PBS or freeze-thawed (F/T) GL261 cells [160]. Importantly, the DC vaccine pulsed with PS-PDT-induced dying cancer cell is proposed as an attractive approach for producing whole-tumor derived tumor-specific antigens (TSA) that may provide superior efficacy compared to conventional F/T cell lysates which often lead to weak immunogenicity and unregulated cell death by accidental necrosis [160,163,193,194]. In contrast to non-mutated self-antigen TAAs, TSAs or neoantigens are recognized as non-self-antigens that are generated by cancer cells as a result of tumor-specific alterations, such as mutation, RNA splicing, and post-translational modification [195]. Several studies have shown that using F/T cells as vaccines failed to generate antigen-specific CD8^+^ T cell-mediated immune responses [163,196,197,198,199].

It is noteworthy that autologous tumour cells possess a full repertoire of personal mutant TSAs including those that could be crucial for tumour rejection, which are absent in allogeneic tumor cell lines [200]. To promote antitumour immunity in a patient-personalized manner, a study developed an in situ DC vaccine that is administered via a single low-dose intravenous (i.v.) injection of an artificial vesicle polymersomal combination along with PDT without the need of loading DCs ex vivo [143]. A chimeric polymersome (CCPS) co-encapsulated a low-dose of DOX and PS. Chemotherapeutic agent DOX and PS-PDT induced ICD by releasing HMBG1 and CRT and forming reactive oxygen species [143,201]. Moreover, CCPS embedded with amine groups served as adjuvants and enhanced DC maturation and cross-presentation abilities by 19-fold compared to negative control group [202]. MC38 colorectal cancer-bearing mice treated with a single i.v. injection of CCPS_DOX_PS-PDT showed the highest serum levels of proinflammatory cytokines, IL-6, IL-12, and TNF-α, and the prominent elimination of primary tumors [143,203,204]. Furthermore, levels of mature migratory DCs and tumor-infiltrating CTLs were elevated in CCPS_DOX_PS-PDT vaccinated mice compared to those vaccinated with CCPS encapsulating DOX or PS alone, suggesting the importance of co-encapsulation of DOX and PS in developing immunogenicity via PDT. In addition, a significant abscopal effect was observed in mice treated with CCPS_DOX_PS-PDT [143]. Such a combination in in situ DC vaccines incorporating synthetic delivery vehicles, chemotherapeutic drugs, and PDT may potentially enhance DC maturation and therapeutic vaccine efficacy.

PTT induces ICD for tumor ablation with the heat converted from near-infrared laser (NIR) absorbed by photothermal agents that are accumulated in the tumor [205]. Recently, combinations of PTT with other therapeutic modalities such as nanomaterials, immunotherapies, or even PDT and conventional chemo- and radiotherapy were demonstrated to provide additive or synergistic therapeutic efficacies [206,207,208,209]. In particular, multiple studies have demonstrated that PTT combined with immunotherapy augmented the immune response by ICD of local cancer cells and the release of TSAs and cytokines, and inhibited metastases [210,211,212]. One study established a combination of PTT with immune-adjuvant nanomaterial that co-encapsulated a photothermal agent, indocyanine green (ICG), and an immune adjuvant TLR7 agonist imiquimod (R837) in a PLG scaffold, forming a PLG-ICG-R837 nanoparticle [213]. Upon NIR-induced PTT on primary tumors injected with PLG-ICG-R837, enhanced levels of DC recruitment into the tumor site, DC maturation in TDLNs, and sera levels of proinflammatory cytokines were observed [213]. Furthermore, it was shown that this combination with ICB established abscopal effects after the ablation of primary tumors and long-term anti-tumor immunogenicity, as secondary tumor growth of 4T1 breast cancer and CT26 colorectal cancer were completely inhibited [213]. Systemic administration of this formulation followed by PTT also offered a strong antitumor response without observable cytokine-storm-like side effects, supporting its efficacy in tumors that can hardly be reached [213]. Synergistic interactions between PTT and other therapeutic modalities, particularly immunotherapy, appear promising to enhance antitumor vaccine efficacy and immunological memory besides minimizing metastases, encouraging further studies [214].

## 6. mRNA-Pulsed DC Vaccines

The pulsing of DCs with tumor antigens is a widely used vaccination strategy in which autologous DCs are loaded with antigenic information and matured under favorable conditions [215]. The resulting DCs are then administered back into the patient to initiate antitumor immune responses [215]. Antigens that are used to pulse DCs are commonly obtained as whole-cell antigens from repeated F/T cancer samples and synthetic cancer antigenic peptides [215,216,217]. Cancer antigenic peptides are used in many clinical trials, but they involve substantial weaknesses [218]. For example, the selected peptides to be synthesized require prior identification of human leukocyte antigen (HLA) types and have HLA restriction of patients [219]. In addition, peptides may only serve to elicit either CD4^+^ or CD8^+^ T cell response, and the short half-life of HLA-antigenic peptide complex limits durable antigen presentation [21,113].

mRNA has emerged as an appealing candidate that offers both antigen delivery and innate immune activation under spatiotemporal control that circumvents the limitations of common tumor antigen sources [220]. mRNA pulsing of DCs is widely recognized due to its multiple advantages: the introduction of exogenous mRNA activates innate immune cells by stimulating various TLRs such as TLR3, TLR7, and TLR8, presenting strong intrinsic adjuvanticity; does not integrate into the genome, avoiding any insertional mutagenesis; can be readily produced in large amounts in vitro; can be engineered to increase immunogenicity and reduce inhibition of its translation; not subject to splicing, eliminating any uncertainty in protein products due to alternative splicing [215,221,222,223,224]. 

Since the feasibility of ex vivo mRNA-pulsed DC vaccine was first demonstrated 27 years ago, numerous preclinical and clinical studies have explored loading DCs with mRNAs as a new vaccination strategy [218,225]. Multiple studies have confirmed the ability of mRNA-pulsed DCs to induce potent tumor antigen-specific T cell responses in vivo [226,227,228,229]. Among several strategies developed to introduce mRNA directly into DC cytosol, electroporation has been demonstrated to be the most efficient method [230,231]. By means of weak electric pulse, mRNA is rapidly integrated into the cytosol, which prevents mRNA from degradation by ubiquitous extracellular ribonucleases [218]. Electroporation also alleviates the potential alteration of DC immunophenotype, maturation potential, migration capacity, and T cell-stimulatory potential [232,233].

DCs may be pulsed with mRNA from either source: whole tumor-derived mRNA or in vitro transcribed mRNA [218,234]. Tumor-derived mRNAs allow the delivery of complete repertoire of epitopes expressed in tumor, which circumvents immune escape due to antigen loss or downregulation [218]. The safety of autologous tumor mRNA-pulsed DC vaccines was demonstrated in clinical trials of various cancer types, including renal cell carcinoma (Phase I, NCT00006431) and advanced malignant melanoma (Phase I/II, NCT01278940) [235,236]. The latter study demonstrated vaccine-elicited T cell responses in about 51% of the patients and significantly enhanced survival in immune respondents [237]. However, relatively large amounts of autologous tumor cells and adequate accessibility of the tumor site are required to prepare whole tumor-derived mRNA [220]. In addition, tumors express many other molecules besides TSAs, such as self-antigens and inhibitory molecules [234]. These irrelevant peptides may become more abundant after a multitude of steps involved in the in vitro amplification of tumor-derived mRNAs and promote the under-representation and loss of immunogenic antigens or epitopes [238].

Therefore, in vitro generated mRNAs are preferentially employed in preparations of DC vaccines [238]. Although the safety profiles of in vitro generated mRNAs have been demonstrated in numerous clinical studies (Phase I/II, NCT00243529; Phase II, NCT00965224, NCT01446731, NCT02285413) [219,239,240,241,242], general clinical efficacies have been low due to the weak induction of T cell responses [220]. Thus, modifications to enhance the expression of mRNA-encoded antigen and translational efficiency and to diminish mRNA extracellular degradation level were performed [243]. Moreover, it has become increasingly clear that CD4^+^ T cells, especially Th1 cells, are important contributors in the antitumor activity on top of CD8^+^ T cells, and that they are essential for the formation of memory CD8^+^ T cells [244,245,246]. However, antigenic peptides that are processed from mRNA-encoded antigenic proteins in the cytosol of DCs and presented onto MHC I are only capable of stimulating CD8^+^ T cells. Since the activation of CD4^+^ T cells depends on MHC II presentation pathway, antigens are targeted to lysosomes by means of fusion to lysosomal sorting signals such as lysosome-associated membrane protein-1 (LAMP-1) or MHC I trafficking signal (MITD) [247,248,249]. Such LAMP-1- or MITD-fused chimeric proteins are processed and presented onto both MHC I and II to expand both CD4^+^ and CD8^+^ T cell responses and improve effector functions [219,250]. Clinical studies investigating the effect of LAMP-1 mRNA-loaded DC vaccines in glioblastoma have observed exceptional tumor-specific CD4^+^ and CD8^+^ T cell response and extended overall patient survival (Phase I, NCT00626483, NCT00639639; Phase II, NCT02366728) [251,252].

To further ameliorate DC vaccine efficacy, strategies that co-transfect DCs with mRNAs encoding immunostimulatory ligands and receptors to enhance DC maturation and T cell co-stimulation have been developed [220]. In preclinical studies, co-transfection with mRNAs encoding CD83, OX40 ligand (OX40L; CD134), 4-1BB ligand (4-1BBL; CD137L), CD40 ligand (CD40L), and glucocorticoid-induced tumor necrosis factor receptor ligand (GITRL) have shown enhanced DC co-stimulation and T cell priming [253,254,255,256,257]. Co-administration of antigenic and 4-1BBL mRNA significantly increased tumor-specific CTL response compared to only antigenic mRNA administration [256]. Another study demonstrated the synergistic effect of 4-1BBL and CD40L mRNA co-stimulation on enhancing both CD4^+^ and CD8^+^ T cell responses and alleviating Treg-suppressed CD8^+^ T cell proliferation [258]. Rocapuldencel-T is currently the most clinically advanced mRNA-pulsed DC vaccine, which integrates the co-transfection of autologous tumor-mRNA and CD40L mRNA [259]. However, it has failed to improve the overall survival of metastatic renal cell carcinoma patients in combination with standard treatment despite its tolerability and capability to increase T effector memory cells (T_EM_) [260]. Therefore, it led to withdrawal or termination of the clinical studies (Phase II, NCT02662634; Phase III, NCT01582672) [259,260]. Of note, TriMix formulation, a cocktail of mRNA-encoded adjuvants (CD70, CD40L, and constitutively active TLR4) that can be electroporated together with antigenic mRNA(s) was developed as an alternative approach [261,262,263]. Importantly, the electroporation of mature DCs with TriMix demonstrated its ability to reprogram Tregs to Th1-like cells with enhanced IL-12 secretion, therefore alleviating Treg inhibition of CD8^+^ T cells in vitro and in vivo [263,264].

The combination of mRNA-pulsed DCs and ICBs has also been investigated to maximize vaccine efficacies. In a single-arm phase II clinical study (NCT01302496), patients with stage III/IV malignant melanoma were treated with DCs, electroporated with TriMix, and mixed with multiple LAMP-1-fused melanoma-associated tumor mRNAs (TriMixDC-MEL), along with ipilimumab (IPI; CTLA-4 inhibitor) mAb [265]. The combination of TriMixDC-MEL and IPI was well tolerated and resulted in durable tumor reduction in patients with recurrent or refractory melanoma [266]. These promising preclinical data of co-transfection with antigenic tumor mRNA and IPI mAb may also encourage ICBs of CTLA-4 and/or PD-1 pathways [218]. Synergistic effects were observed in the co-transfection with mRNAs encoding anti-CTLA-4 and anti-PD-1 mAbs, which were associated with higher overall response rates and greater changes in tumor burden compared to single agent treatment, conferring it as the first FDA-approved treatment for advanced-staged melanoma [267,268]. However, the association of anti-CTLA-4 mAb with severe adverse effects and the superior efficacy and safety of anti-PD-1 mAb over IPI are encouraging more studies to focus on the combination of TriMixDC-MEL with anti-PD-1 treatment [269,270,271,272].

Co-transfection of DCs with mRNAs encoding stimulatory cytokines was also employed to enhance DC maturation and T-cell priming in autologous DC vaccines loaded with antigenic mRNA. Several stimulatory cytokines were explored, including GM-CSF, IL-12, and IL-15 [273,274,275,276,277]. IL-12 is a potent cytokine that mediates the differentiation of Th1 cells and stimulates the cytotoxic abilities of natural killer (NK) cells and CTLs [278,279]. However, electroporation of IL-12 mRNA alone into DCs did not affect cell survival or maturation status [280]. Bcl-2 is a critical pro-survival factor that exerts anti-apoptotic functions [281,282]. One study demonstrated the synergistic effects of electroporation of DCs with IL-12 and Bcl-2 mRNAs on priming CD8^+^ T cells and DC viability to assist vaccine potency and clinical efficacy [280]. In addition, an ongoing phase I clinical trial, in which IL-12 mRNA is administered intratumorally in combination with anti-PD-L1 mAb durvalumab in patients with advanced solid tumors, has shown the safety and tolerability of such combinational therapy (NCT03946800) [283,284].

In addition to ex vivo electroporation, several in vivo delivery methods to pulse DCs using “naked” mRNA have been developed because the administration route and delivery format considerably determines immune response [285]. For example, i.t. delivery of TriMix induced the maturation and migration of tumor infiltrating DCs toward TDLNs, stimulating antitumor responses against spontaneously acquired mRNAs [262,286,287]. The curative potential of i.t. delivery of TriMix is suggested to exploit autologous tumor antigenic repertoire without the need of tumor antigen co-delivery, and to stimulate CTLs against various tumor antigens in situ [220,287,288]. A phase I study of i.t. delivery of TriMix in patients with early breast cancer lesions is ongoing (NCT03788083) [287]. Intranodal (i.n.) injection delivers mRNA directly into the secondary lymphoid structure, which offers the advantage of targeting antigen delivery at the site of T cell activation, obviating the need for DC migration, and selective uptake by DCs to elicit prophylactic or therapeutic antitumor responses [285,289]. Simultaneous i.n. co-delivery of TriMix and antigenic mRNA recruited antigen-specific CD4^+^ and CD8^+^ T cells, and CTLs against various TAAs [264]. Clinical trials have also demonstrated the safety and tolerability of i.n. administration of TriMix combined with mRNAs, encoding melanoma-specific TAAs in patients with resected melanoma (Phase I, NCT03394937) [290,291]. Intradermal (i.d.) administration offers an ideal site of delivery, as various APCs reside in the skin [285,292], and the efficacies of i.d. delivery of self-adjuvanted mRNA-encoded TAAs have been established in various mouse cancer models and clinical studies [293,294,295]. On the other hand, i.v. administration of mRNA vaccines was concerned with rapid degradation of mRNAs by ribonucleases and poor target expression in secondary lymphoid structures [285,288]. An mRNA–liposome complex (mRNA-lipoplex) platform was generated to protect mRNA integrity and deliver mRNAs to target DCs in lymph nodes based on the net charge of the platform particles upon i.v. injection [13,296,297]. Since the safety of mRNA-lipoplex was demonstrated in animal models, several clinical trials were initiated (Phase I, NCT02410733, NCT02316457) [298,299]. Although numerous studies have shown efficacies of different mRNA-pulsed DC vaccine administration strategies, it has yet to come to a consensus as to which elicits a superior prophylactic or therapeutic response [218]. Recently, combining multiple routes has been proposed to induce a more systemic immune response in clinical trials of i.d. and i.v. administration of TriMix-MEL as a single agent (Phase Ib, NCT01066390; Phase II, NCT01676779) and in combination with IPI (Phase II, NCT01302496) [265,300,301,302].

## 7. DC-Derived Small Extracellular Vesicle (sEV) Vaccines

The major limitations of using conventional MoDCs to prepare DC vaccines are the immunosuppression in the TME, off-target toxicities, and the need for autologous cells [303]. To address such challenges encountered by conventional DC-based immunotherapies, cell-free DC-based vaccines incorporating DC-derived small extracellular vesicles (DCsEV) which are more resistant to immunosuppression are proposed [304]. Extracellular vesicles (EVs) are lipid-bound vesicles released by all cellular organisms and play key roles in intercellular crosstalk via the delivery of EV cargo, cell surface modifications, and target-cell modulation [305]. Small EVs (sEVs) are one of the heterogenous subsets of EVs, characterized by their size, and are of 30–150 nm or less than 200 nm in diameter [306]. EVs contain diverse cargo, including DNA, mRNA, non-coding RNAs, proteins, and lipids [307]. DCs process exogenous antigens in endosomal compartments that can fuse with the plasma membrane to release inert sEVs. DCsEVs play a significant role in DC-to-cell communications via transferring cargos that contain proteins, metabolites, and nucleic acids [304,308]. They carry functional surface antigenic pMHC I and pMHC II, as well as co-stimulatory molecules (CD80/86), and are thus capable of priming antigen-specific CD8^+^ T cells [309]. Of note, DCsEVs present more pMHC complexes, especially 10 to 100 folds greater pMHC II than DCs [304,310]. Additionally, they are highly biocompatible [309]. While maintaining the essential immunostimulatory properties of DCs, DcsEVs allow for frozen storage and are more amenable to strict vaccine manufacturing processes, and lack the risks associated with viable cellular or viral therapies such as the risk of in vivo replication [304,311].

Despite the strong safety profiles of DCsEVs as cancer immunotherapy, previous clinical trials of first-generation peptide-loaded DCsEVs from autologous MoDCs showed limited efficacies (Phase II, NCT01159288) [312,313,314]. Such a limited immunizing capacity may be due to the insufficient antigen presentation, elevation of Treg level, unknown efficiency of in vivo DCsEV trafficking, and surface expression of immunoregulatory molecules such as PD-L1 [312,313,314]. The second generation of DCsEVs were derived from IFN-ɣ-matured MoDCs [312,315]. Notably, the maturation status of donor DCs influenced the function of DCsEVs and shaped opposing immune responses [304,316]. DCsEVs derived from mature, stimulated MoDCs promoted the exchange of functional pMHC complexes with DCs and enhanced antigen-specific T cell responses, while DCsEVs constitutively released from immature DCs advanced T cell tolerance, Treg proliferation, and the suppression of CD4^+^ T-helper 17 cells and their production of immunostimulatory cytokine IL-17 [317,318]. IFN-ɣ-matured MoDCs generated DCsEVs that expressed elevated levels of pMHC II, costimulatory molecules, and intercellular adhesion molecules (ICAMs; important for DC homing to LNs), which thereby enhanced the immunostimulatory properties of DCsEVs [304,317]. However, the antigen-specific T cell immune response of the second generation of DCsEVs was still impeded, likely due to insufficient antigen presentation ability and poor in vivo distribution [304]. Several clinical trials on the basis of autologous DCsEVs are in the developing stages, as the safety of DCsEVs have been proved in clinical trials and their prospects in cancer therapy remain promising.

Many studies focus on peptide- or protein-loaded DCsEVs from MoDCs. Other types of DCsEVs that are capable of priming CD8^+^ T cell responses in vivo are needed to expand the studies of DCsEVs. Despite the presumed tolerogenic role of pDCs in tumors, several clinical trials have shown that vaccination with pDCs possibly leads to the induction of antitumor CD8^+^ T cell immunity [319,320]. Recently, a new subset of DCsEVs was discovered as the cross-presenting pDCs required bystander cDCs to cross-prime CD8^+^ T cells in vitro and in vivo by transferring antigens via pDC-derived small EVs (pDCsEVs) [321]. Therefore, pDCsEVs are suggested as a mean to integrate both cDCs and pDCs in cancer vaccines to achieve better anti-tumor efficacy [319]. Although further studies are needed to investigate whether pDCsEVs generated with pDC-targeted antigens exhibit enhanced cross-presentation capacity similar to DCs targeted with antigens and are capable of priming allogeneic CD8^+^ T cell response similar to protein-loaded DCsEVs, a new subset of DCsEVs suggests a significant mechanism to enhance T cell cross-priming and induce antitumor efficacy. While DCsEVs can potentially overcome the limitations of MoDC-based vaccine, further investigations are needed to elucidate the underlying mechanisms between DCsEVs and CD8^+^ T cell priming in vivo to advance their clinical application [68].

Recently, a third generation of DCsEV-based cancer vaccines has been developed using nanotechnology and molecular engineering [322]. DCsEVs are engineered to amplify certain immunostimulatory characteristics [304]. A novel DCsEV engineering approach has formed a “designer” vaccine as a universal immunotherapeutic strategy for hepatocellular carcinoma (HCC) [309]. These engineered DCsEVs present an HCC-targeting peptide P47, an antigenic epitope AFP212-A2, and a functional domain of high mobility group nucleosome-binding domain (HMGN1; N1ND) as an immunoadjuvant to recruit and activate DCs [309,323]. These engineered DCsEVs were i.v. injected and successfully enabled tumor-targeted N1ND-mediated recruitment and activation of endogenous cross-presenting DCs in the tumor of orthotopic HCC mice [309]. Moreover, antigen-specific induction of tumor-specific T cell responses and tumor suppression was achieved, supported by the elevated levels of IFN-ɣ-expressing CD8^+^ T cell and decreased levels of immunosuppressive cytokines such as IL-10 and TGF-β [324]. This study highlighted the capacity of engineered DCsEVs to induce tumor-specific immune response and to be presented as a generalizable approach as a personalized immunotherapy without having to identify patient-specific TSAs [309].

In addition to engineering immunostimulatory activities of DCsEVs, the incorporation of ICBs has been developed. A combination of DCsEVs and ICBs (anti-CTLA-4 therapy or PD-1/PD-L1 blockade) may further enhance the cross-priming of CD8^+^ T cells and alleviate the suppression of tumor-infiltrating CTLs [304,325]. For instance, anti-CTLA-4 mAb were anchored to the surface of DCsEVs derived from ovalbumin (OVA) antigen-pulsed, activated, and TLR-3 agonist poly(I:C)-matured immature DCs. Poly(I:C) is also a TLR-3 agonist from which its derivative poly-ICLC is generated [326]. This approach generated a bifunctional CTLA-4-mAb-modified DCsEVs (DCsEV-OVA-mAb) [327,328]. Poly(I:C) was demonstrated to effectively activate DCs, augment cross-presentation to CD4^+^ and CD8^+^ T cells, and enhance antitumor response in cervical cancer and melanoma immunotherapies. These poly(I:C)-matured DCs secreted DCsEVs with similar enhanced antitumor functions [329]. These functionalized DCsEV-OVA-mAb were enriched in MHC I and II and co-stimulatory molecule CD80. Compared to DCsEVs pulsed with OVA without mAb engineering, DCsEV-OVA-mAb induced a stronger T cell activation and proliferation in vivo, highlighting the importance of CTLA-4 mAb in the vaccine efficacy [327]. They also quickly migrated to TDLNs upon s.c. injection, increased i.t. CD4^+^ and CD8^+^ T cell migration, and significantly increased CTLs/Treg ratio in tumor [327]. Importantly, vaccination with DCsEV-OVA-mAb generated the most T_EM_ cells, in line with previous studies that showed enhanced generation and function of CD8^+^ T_EM_ cells post-CTLA-4 blockade therapy [330,331,332]. Altogether, surface modification of DCsEV with the incorporation of anti-CTLA-4 mAb provides a promising strategy to induce potent antitumor efficacy of DCsEV vaccination against cancer [327].

Moreover, a platform of genetically engineered DCsEVs incorporated with anti-PD-1 antibody on the surface which allows a direct presentation of tumor antigens to CD8^+^ T cells and stimulate strong CTL responses has been developed [333]. An immature DC2.4 cell line was engineered to display anti-PD-1-single-chain antibody fragment on their membrane surface. These α-PD-1-DCs were then activated and matured by recombinant adenovirus transduction of melanoma TSAs. The matured DCs were upregulated in the expressions of pMHC I, anti-PD-1 antibody, costimulatory molecules (CD80/86), and ICAM-I. The resultant DCsEVs collected and purified from these modified DCs were named ASPIRE [333]. Importantly, ASPIRE expressed the same immunostimulatory molecules as the engineered DCs. ASPIRE stimulated antitumoral CTL response in mice via direct transfer of pMHC I complex to endogenous DCs instead of intermediate DCsEV uptake by DCs, contrary to the mechanisms proposed by earlier studies [304,334,335]. When ASPIRE was inoculated into B16-F10 melanoma tumor-bearing mice, complete tumor rejection was achieved in all mice by successful migration of ASPIRE to TDLNs, where it contacted and activated CTLs. In addition, ASPIRE resulted in less PD-1^+^ dysfunctional CTLs in tumor-bearing mice than a combinatory treatment of unmodified antigen-presenting DCsEVs along with a free PD-1 mAb. The potential clinical translation of ASPIRE needs further investigation as it is currently unknown whether ASPIRE preparation can also be implemented in standard source of human autologous MoDCs for clinical use and whether it will establish similar immunostimulatory properties as the murine cell line preparation [336]. Nonetheless, ASPIRE presents a novel therapeutic vaccine platform to stimulate enhanced CTL immune response and overcome the immunosuppressive TME by the incorporation of anti-PD-1 ICB in DCsEV vaccines. 

To develop novel cancer vaccines using DCsEVs, which is a highly heterogenous population that is often extremely difficult to obtain from a specific immune cell population without contamination with sEVs from other cell types, detailed characterization and careful collection of the sEVs are critical [306]. Recent progress in molecular engineering and the delivery of immune modulatory adjuvants have indeed promoted the antitumor response of DCsEV-based vaccines. Further understanding of sEVs, especially in their biogenesis from immune cells and interactions with tumor and other cell types as well as in vivo biodistribution tracking of them, are deemed as providing a significant breakthrough in sEV-based therapeutics. 

## 8. Tumor-Derived sEV Vaccines

In addition to DCsEVs, tumor-derived sEVs (Tu-sEV) are proposed to be utilized as a source of tumor antigens to develop cancer vaccines [337]. Tu-sEVs share several similar functional characteristics with DCsEVs, but they shuttle information between tumor cells and the TME and are highly enriched in parental tumor antigens [338,339]. They also express distinct sets of proteins that facilitate their binding to and uptake by DCs, such as MHC, LAMP-1, CD9, and CD54 [340,341,342,343]. They can therefore stimulate a broad range of tumor-specific CTL responses against multiple antigenic epitopes [339,344]. Moreover, Tu-sEVs are easily isolated and purified by non- or minimally-invasive methods from patient’s plasma, ascites, and pleural effusions [338]. Tu-sEVs also have significant advantages such as antigen sources, attributed to their high immunogenicity from stressed tumor cells, more than irradiated tumor cell lysates [345,346]. Successful uptake and processing of Tu-sEVs by DCs can enhance the expression levels of co-stimulatory molecules, MHC, and CD11c, and lead to phenotypically and functionally mature DCs [347,348,349]. These features altogether support Tu-sEV as an attractive cell-free cancer vaccine candidate in a patient-personalized manner [350,351,352,353].

However, it is important to note that Tu-sEVs regulate immune responses through both immunosuppressive and immunostimulatory functions, promoting either immune escape or tumor regression [354,355,356]. Specifically, Tu-sEVs can induce the production of inhibitory cytokines, decrease the expression of co-stimulatory molecules, increase STAT3 expression, modulate DC differentiation, and inhibit maturation and the T cell stimulatory capacity of DCs [354,357,358]. Tu-sEVs impair DC maturation and antigen-specific responses via the downregulation of MHC II expression as well as the expansion of Tregs [357,359,360,361]. Earlier studies have, however, proven the feasibility and efficacy of Tu-sEVs as prophylactic and therapeutic cancer vaccines that not only elicit CTL responses but also prevent metastases in mouse models [339,345,351,362,363]. Meanwhile, possible Tu-sEV-derived immunosuppression has redirected further investigations of standalone Tu-sEV treatments to the development of Tu-sEV-engineering and DC loading to advance the immunogenicity of Tu-sEV-based cancer vaccines [337].

Cancer cells or Tu-sEVs themselves may be manipulated to increase the expression levels of tumor antigens or immunostimulatory molecules that are expressed on Tu-sEVs [337]. This may be achieved by transducing tumor cells with viral vectors transfected with genes encoding immunogenic antigens, such as mucin 1 (MUC1), which is a transmembrane glycoprotein that is overexpressed in many cancers, including prostate, breast, and ovarian cancer [364,365,366,367]. MUC1-transduced CT26 colon and TA3HA breast murine cancer cell lines secreted Tu-sEVs that successfully expressed target antigen MUC1 (MUC1-Tu-sEVs) [368]. Both autologous and allogenic MUC1-Tu-sEVs stimulated immune responses and inhibited tumor growth 2-fold greater than control Tu-sEVs, independent of their MHC types in vivo [368]. Furthermore, MUC1-Tu-sEVs activated splenocytes via DCs in a MUC1-dependent manner, inducing the secretion of Th1-type IFN-ɣ in vitro [368]. Recently, another study has directly anchored IFN-ɣ fusion protein onto the surface of RM-1 prostate cancer cell-derived Tu-sEVs (IFN-ɣ-Tu-sEVs) [369]. Vaccination with these IFN-ɣ-Tu-sEVs resulted in the highest levels of CD4^+^, CD8^+^, and IFN-ɣ^+^ CD8^+^ T cells [369]. Importantly, they decreased the levels of Tregs and PD-L1, and IDO expressions in the TME [369]. IFN-ɣ-Tu-sEVs also significantly inhibited tumor growth and prolonged the survival time of tumor-bearing mice [369]. Furthermore, another study incorporated interferon regulatory factor 1 (IRF-1), which is a tumor suppressor gene that regulates the expression of target genes such as MHC I, IL-15, and IFN-α [370]. Of note, IRF-1-Tu-sEVs displayed an enhanced antitumor response compared to IFN-ɣ-Tu-sEVs [370]. This functional response was mediated through elevated expressions of MHC I and IL-15Rα, resulting in increased tumor infiltrating CD4^+^ and CD8^+^ T cells [370]. These studies altogether support cancer cells or Tu-sEVs may be genetically engineered to enhance antitumor responses of Tu-sEV-based vaccines via elevated levels of immunostimulatory antigen and molecule expressions as well as both CD4^+^ and CD8^+^ T cells.

In addition to immunogenic tumor antigens, MHC II is another important molecule that could increase the immunogenicity of Tu-sEV-based vaccines [371]. Many cancer immunotherapies have focused on inducing MHCI I-restricted tumor-specific CTL responses because most tumor cells constitutively express MHC I, but not MHC II [372,373]. However, to optimally induce both humoral and cellular effector mechanisms, MHC II-restricted CD4^+^ T cells that support the maturation, proliferation, and functionality of CD8^+^ CTLs are also required [374]. Hence, a study transduced B16-F1 murine melanoma cells with MHC class II transcription activator (CIITA) gene to generate Tu-sEVs enriched in MHC II and tumor antigen TRP2 (CIITA-Tu-sEV) [375]. Compared to parental control Tu-sEVs, CIITA-Tu-sEVs exhibited greater DC maturation ability, which was reflected by higher MHC II and CD86 expression, and higher mRNA levels of inflammatory cytokine, TNF-α, maturation marker, CCR-7, and Th1-polarizing cytokine, IL-12 in vitro [375]. CIITA-Tu-sEVs also improved the preventative and therapeutic antitumor immune response and increased Th1 type antibody and IFN-ɣ levels in vivo in a dose-dependent manner [375]. Moreover, CIITA-Tu-sEVs exerted greater tumor rejection, increased survival rate, and overall survival by 20% at 60 days post-treatment [375]. Taken together, MHC II-expressing Tu-sEVs may be another potential candidate to be utilized in cancer vaccines to augment both cellular and humoral antitumor immune responses.

Engineered Tu-sEVs and vaccine adjuvant co-delivery system is also proposed to exhibit potent antitumor activity and to form an effective in situ DC vaccine. Murine melanoma B16-BL6 cells were transfected with a plasmid vector that encoded a fusion streptavidin (SAV)-lactadherin (LA) protein, which yielded genetically engineered SAV-Tu-sEVs [376]. SAV is a protein that binds to biotin with high affinity, and LA is an sEV-tropic protein that binds to the EV membrane [377,378]. SAV-Tu-sEVs were then incubated with biotinylated CpG-ODN to prepare CpG-ODN-modified SAV-Tu-sEVs (CpG-SAV-Tu-sEVs) [376]. These modified CpG-SAV-Tu-sEVs successfully delivered CpG-ODN to DCs, and activated and enhanced tumor antigen presentation abilities of DCs in vitro. Significantly increased levels of cytokine, such as TNF-α, IL-6, and IL-12p4, were also observed from CpG-SAV-Tu-sEV-pulsed DCs [376]. Moreover, immunization with these CpG-SAV-Tu-sEVs resulted in potent cellular and humoral immunity along with upregulation of Th1-related antibody as well as protective and therapeutic antitumor immunity [376]. CpG-SAV-Tu-sEVs demonstrated stronger in vivo antitumor effects in B16-BL6 tumor-bearing mice than separate administrations of CpG-ODN and Tu-sEV, suggesting the co-delivery of tumor antigens and adjuvants by genetically engineered Tu-sEVs as a potential immunotherapy approach.

The notion of Tu-sEV-mediated adjuvant delivery was supported by another study that evaluated the immunogenicity of mouse breast cancer cell-derived Tu-sEVs which were loaded with two immunoadjuvants, CpG-ODN and poly(I:C) (CpG-p(I:C)-Tu-sEV) [379,380]. These engineered CpG-p(I:C)-Tu-sEVs exhibited augmented immunostimulatory properties by activating antigen-specific primary and memory T cell responses and promoting tumor regression in tumor-bearing mice [379]. Similar to the Tu-sEV-mediated antigen-adjuvant co-delivery system above, these also elicited Th1-biased immunity reflected by elevated levels of Th1-related antibody and IFN-ɣ [379]. Altogether, this study also showed that Tu-sEV-based therapeutic vaccine can promote strong cellular and humoral antitumor immunity that can provide a personalized tumor therapy strategy.

In addition to genetic engineering of Tu-sEVs, their surface may also be modified to present immunoadjuvant. One study painted the surface of Tu-sEVs with the functional domain of HMGN1 (N1ND-Tu-sEV) via a vesicular anchor peptide [381]. DCs pulsed with these N1ND-Tu-sEVs-pulsed were activated and boosted CD8^+^ T cell levels. N1ND-Tu-sEVs also significantly increased DC migratory capacity and the generation and amplification of T_EM_, which contributed to long-lasting antitumor immunity and tumor suppression in different syngeneic immunogenic mouse models with large tumor burdens, most notably in low immunogenic orthotopic HCC [382]. Importantly, N1ND-painted serum sEVs from cancer patients could also induce tumor-specific cytolysis in vitro and promote DC activation [376]. This study demonstrated the potency of surface-modified Tu-sEVs to augment DC immunogenicity and to inhibit large established tumors, thus providing a platform to load immunoadjuvants onto Tu-sEVs to amplify the antitumor immunity of DC vaccines.

Tu-sEVs may also co-deliver ICD inducers and adjuvant to generate an in situ vaccine. One study loaded an ICD inducer, human neutrophil elastase (ELANE), and a TLR3 agonist Hiltonol onto breast cancer-derived Tu-sEVs to form an in situ vaccine (HELA-sEVs) [382]. These engineered HELA-sEVs were also enriched in breast-specific protein α-lactalbumin (α-LA), which granted them an enhanced targeting capability to specifically induce ICD in breast cancer cells [382]. HELA-sEV-induced ICD of breast cancer cells followed by TLR3 adjuvant-induced i.t. accumulation of cDC1s and cross-primed immunogenic CD8^+^ T cell responses. This significantly inhibited the tumor growth in a poorly immunogenic breast cancer mouse xenograft model and patient-derived tumor organoids [382]. Tu-sEV engineering strategies that allow the co-delivery of ICD inducer and adjuvant to trigger in situ ICD and cross-priming are promising for generating an effective in situ DC vaccine and may be extended to other types of solid cancers.

In the meantime, it is critical to carefully investigate the possible adverse roles of Tu-sEVs in impairing differentiation, maturation, and functions of DCs to harness their immunostimulatory capacity while avoiding immune escape [350,383]. Several mechanisms underlying the immune inhibitory effects of Tu-sEVs have been proven and more are being postulated, such as the interaction between Tu-sEV, PD-L1, and DC PD-1, reduced antigen sensing and costimulatory molecule expressions in DCs, and the production of VEGF and IL-10 to inhibit DC differentiation maturation [355,384,385,386,387,388,389]. Although numerous preclinical data support Tu-sEV as a potential source for cancer vaccine, and therapeutic interventions to harness Tu-sEV to stimulate antitumor response are being investigated, significant challenges remain to reject the opposing notion of Tu-sEVs as potential immune suppressors [383]. 

## 9. Conclusions and Perspectives

Cancer vaccines have emerged as an important breakthrough in solid cancer immunotherapy, supported by their safety and promising clinical potential. Despite being employed in most clinical trials, a MoDC-based conventional DC vaccine regimen has shown limited efficacies. In this review, several novel vaccine approaches that are capable of addressing the limitations of conventional DC vaccine have been discussed. The biomaterial-based and ICD-inducing DC vaccines have unique advantages to recruit and activate endogenous DCs, which may be combined to achieve synergistic effects to spatiotemporally provide immunoadjuvants and antigens in situ. In addition, in vitro generated adjuvant and antigen mRNAs are loaded onto DCs to enhance DC immunogenicity. On top of cell-based cancer vaccines, DC- and tumor-derived sEVs are also found to have comparable, if not greater, immunogenicity and induce promising antitumor responses, although further understanding of sEVs is needed. In addition to the discussed novel DC-based therapeutic approaches, combinations with other immunotherapies, including ICBs, recapitulation of heterogenous DC populations, DC maturation with various adjuvants, and different modes of vaccine delivery should also be considered. Despite significant advantages of the selected novel DC vaccines over the conventional DC vaccine, their clinical implications may require further improvements. Multimodal therapies are being robustly investigated to address some of the limitations of monotherapies in Table 1. For instance, limitations of ICD-inducing DC vaccines have been overcome by combinatory approaches with other ICD-inducing and biomaterial-based DC vaccines, as well as ICBs. In addition, biomimetic nanoparticles that can reverse the local immunosuppressive TME have been reported to exert synergistic antitumor responses with the new generations of DC vaccines [390,391,392]. Furthermore, recent advances in next generation sequencing-based patient-specific mutanome mapping have allowed for the identification of the entirety of somatic cancer mutations [393]. Such technology development and further understanding of DC immunogenicity are envisioned to further alleviate immune escape and induce effective antitumor immune responses, which are being investigated in several clinical trials with high frequency of T cell responses reported (Phase I, NCT02035956, NCT03289962, NCT02316457) [394,395,396]. Although much work remains to be done to achieve optimal universal or personalized DC-based immunotherapy, recent significant advances in novel DC vaccine regimen and upcoming clinical trials are expected to encourage therapeutic implementations of DC-based vaccines in the future. 

## Figures and Tables

**Figure 1 cells-12-02147-f001:**
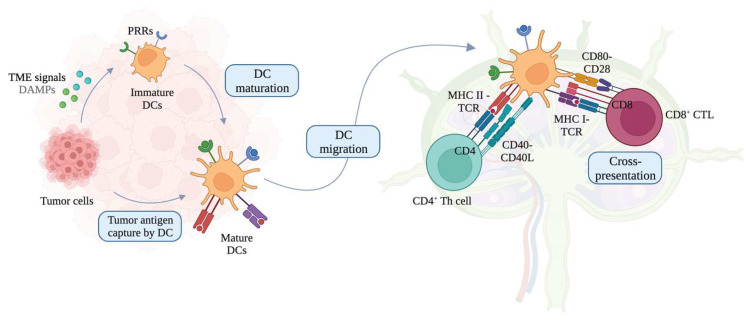
Schematic diagram of tumor antigen cross-presentation by DCs to T cells in TDLNs. DAMPs from tumor cells activate DCs by binding to PRRs in the TME. Mature DCs capture tumor antigens and migrate to TDLNs to perform cross-presentation by presenting acquired tumor antigens on MHC I molecules. Activated, mature, antigen-loaded DCs then cross-prime naïve T cells into antigen-specific T_EFF_ cells including CD4^+^ Th cells and CD8^+^ CTLs via TCRs. Cross-presentation allows the activation and cross-priming of CTLs to induce effective antitumor responses. DC: dendritic cell; TDLNs: tumor draining lymph nodes; DAMP: damage-associated molecular patterns; PRRs: pattern recognition receptors; TME: tumor microenvironment; MHC I: major histocompatibility complex I; MHC II: major histocompatibility complex II; T_EFF_: effector T cells; CD4^+^ Th cells: CD4^+^ T helper cells; CD8^+^ CTLs: CD8^+^ cytotoxic T lymphocytes; TCR: T cell receptor.

**Figure 2 cells-12-02147-f002:**
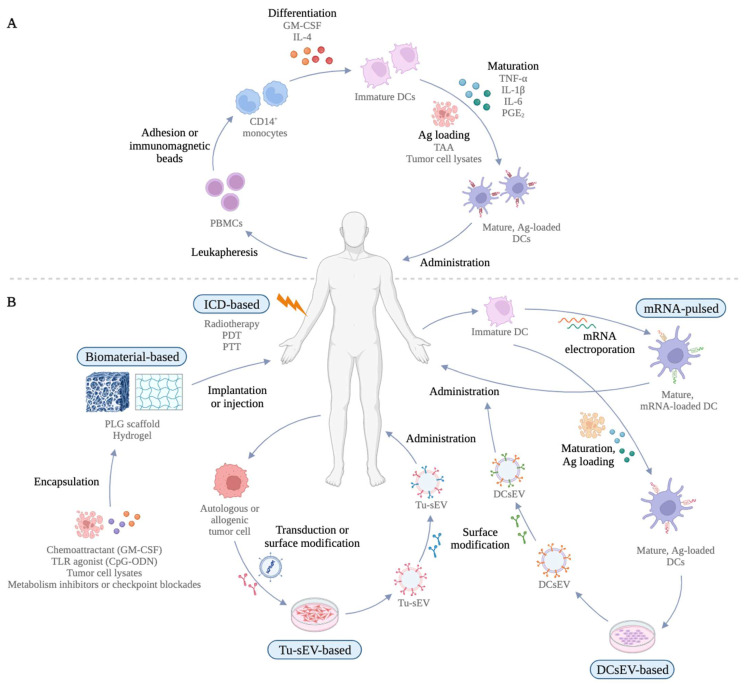
Schematic diagram of conventional ex vivo MoDC-derived DC vaccine and new generations of DC vaccines. (**A**) Patient’s PBMCs are obtained by leukapheresis and CD14^+^ monocytes are selected by plastic adherence or immunomagnetic beads. Monocytes are differentiated into immature DCs in the presence of GM-CSF and IL-4 on day 6. Immature DCs are then matured in a maturation cocktail consisted of TNF-α, IL-1β, IL-6, and PGE_2_, while being simultaneously pulsed with autologous tumor cell lysates or antigens on day 7 for 48 hours. The mature, antigen-loaded DCs are reinfused to the patient. (**B**) Biomaterial-based DC vaccine uses PLG scaffold and hydrogel, which can encapsulate chemoattractant, TLR agonist, tumor cell lysates, and/or metabolism inhibitors or checkpoint inhibitors. PLG scaffold and hydrogel can be implanted or injected, respectively, into the patient to recruit, mature, and activate endogenous DCs in situ. ICD-based DC vaccines use radiotherapy, PDT, or PTT to generate an in situ DC vaccine, which may be facilitated by other adjuvants, immunotherapies, and/or chemotherapy, and combined with biomaterials to enhance the vaccine-induced immune response. mRNA-pulsed DC vaccines involve the electroporation of immature DCs with mRNAs that encode tumor antigens, adjuvants, immunostimulatory ligands, and/or cytokines. DCsEV-based vaccines also make use of immature DCs, which are matured and antigen-loaded with maturation cocktail and tumor peptides or proteins. sEVs secreted from matured, antigen-loaded DCs are collected and undergo surface modifications to enhance their immunogenicity and are injected to the patient. Tu-sEV-based DC vaccines manipulate cancer cells and/or Tu-sEVs to increase the levels of cross-presentation by Tu-sEVs. This may be achieved by transducing tumor cells with viral vectors transfected with genes encoding immunogenic antigens, performing surface modifications, and/or directly modifying Tu-sEV surface. DC: dendritic cell; MoDC: monocyte-derived DCs; PMBC: peripheral blood mononuclear cells; GM-CSF: granulocyte-macrophage colony-stimulating factor; IL-4: interleukin 4; Ag: antigen; TNF-α: tumor necrosis factor α; PGE_2_: prostaglandin E2; PLG: poly(lactide-coglycolide); TLR: toll-like receptor; CpG-ODN: CpG oligodeoxynucleotides; ICD: immunogenic cell death; PDT: photodynamic therapy; PTT: photothermal therapy; sEV: small extracellular vesicle; DCsEV: DC-derived sEV: Tu-sEV: tumor-derived sEV.

**Table 1 cells-12-02147-t001:** Advantages and limitations of conventional MoDC-based vaccine and new generations of DC vaccines.

DC Vaccine Type	Advantages	Limitations	Reference
Conventional MoDC-based	Biocompatibility and safety well demonstrated in multiple clinical trials Satisfactory efficacy seen in phase III clinical trial of Provenge (4.1-month improvement in median survival, 8.7% increased 36-month survival probability in men with metastatic prostate cancer)	Immunosuppressive TME hinders MoDC-induced antitumor responses and migration No standardized DC preparation Functional differences between ex vivo generated MoDC and endogenous DCs Unresponsive in cold tumors Requires leukapheresis Expensive, labor intensive	NCT01832870 (Phase I); NCT01804465, NCT01881867 (Phase II); NCT00065442 (Phase III) [56,57,58,59]
Biomaterial-based	Prone to surface modification to enhance characteristics, biocompatible, biodegradable, and FDA-approved Allows spatiotemporal control of immunostimulatory microenvironment to recruit and activate endogenous heterogenous DC subsets in situ (recapitulation of broad DC responses) Dual role as a sustained vaccine carrier and biomimetic platform Can be combined with ICBs and ICD-inducing therapies to overcome immunosuppressive TME Can inhibit other immunosuppressor cells (MDSCs, Tregs, macrophages) In situ implantation avoids systemic toxicity	Some ex vivo fabricated 3D scaffolds require either surgery or large invasive needle for implantation May require multiple doses Further clinical studies required	NCT01753089 (Phase I) [89,94,101,103,114,115,123,128,131,132,133,397]
Combinatory ICD-inducing	Can convert cold tumors into hot tumors Can generate abscopal effects; i.t. injection with lower dose may also lead to effective treatment and less systemic toxicity Can be combined with ICBs, bio- and nano-materials, and other ICD-inducing therapies to achieve additive or synergistic antitumor responses Can generate whole-tumor derived TSAs to elicit greater immunogenicity in situ Minimally invasive and highly specific	Standalone ICD-inducing vaccine often insufficient to overcome the immunosuppressive TME; require combination with other therapies (ICD-inducing, biomaterial-based, and/or immunotherapies) and additional immunostimulatory adjuvants and cytokines PDT and PTT efficacy may be hindered by deep-seated and large established tumors due to physical and biological barriers	NCT01976585, NCT03789097, NCT00185965, NCT00323882, NCT02221739, (Phase I/II); NCT00880581 (Phase II); NCT00861614 (Phase III) [143,160,181,182,183,184,185,187,202,208,211,213]
mRNA-based	No HLA restriction Introduction of exogenous mRNA stimulates various TLRs; strong intrinsic adjuvanticity Does not integrate into genome; avoiding insertional mutagenesis Can be readily produced in large amounts Can be engineered to increase immunogenicity and mRNA-encoded antigen expression efficiency, and to avoid degradation Not subject to splicing; certain protein products Co-transfection of antigen- and immunostimulatory molecule-encoding (ICB mAbs or cytokines) mRNAs Safety demonstrated in clinical trials	Ex vivo pulsing of MoDCs Rocapuldencel-T as the most clinically advanced autologous tumor-mRNA-pulsed DC vaccine did not improve overall survival of metastatic renal cell carcinoma patients; TriMix developed as an alternative No standard administration/injection method Naked mRNA delivery in vivo may be concerned with rapid degradation and poor target expression in secondary lymphoid structures	NCT00626483, NCT00639639, NCT03946800, NCT03788083, NCT03394937, NCT02410733, NCT02316457 (Phase I); NCT01066390 (Phase Ib); NCT02366728, NCT01302496, NCT01676779 (Phase II) [248,252,256,258,263,265,266,280,283,284,287,289,291,298,299,302]
DCsEV-based	Cell-free vaccine; more resistant to immunosuppression by TME 10–100 folds greater pMHC II than DCs Allows frozen storage, lack risks associated with viable cellular or viral therapies Can be engineered to amplify immunostimulatory characteristics Can be combined with ICBs to enhance antitumor immune response	Some approaches utilize autologous ex vivo manipulated MoDCs The utilization of allogenic DC cell line needs further clinical investigation Immune modulation dependent on maturation status of donor DCs; must be mature and stimulated to elicit antitumor immune response	NCT01159288 (Phase II) [309,312,313,314,315,318,321,327,333]
Tu-sEV-based	Cell-free vaccine; more resistant to immunosuppression by TME Stimulate a broad range of tumor-specific CTL response against multiple antigenic epitopes Easy isolated and purified by non- or minimally-invasive methods High immunogenicity; more than irradiated tumor cells or tumor lysates Can be manipulated by genetic or surface modifications Can co-deliver ICD inducers and adjuvants	Possible Tu-sEV-derived immunosuppression; can induce the production of inhibitory cytokines, decrease co-stimulatory molecule expression, and impair DC maturation and immunogenic functions	[368,369,370,376,379,381,382]

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
