# Peer review of "Dendritic Cell Vaccines: A Shift from Conventional Approach to New Generations"

_cells, 2023, doi:10.3390/cells12172147_

Round 1
Reviewer 1 Report
The review focuses on DC vaccines with a special emphasis on the therapeutic potential of new generation of cancer treatment approaches as alternative for the classical DC vaccine and its limitation. The review is well-written and covers most of new Cancer therapeutic with an emphasis how immunogenic DCs could be enhanced for an optimal cancer treatment. In term of originality, this reviewer covers serval aspect of cancer vaccine that have been published previously. Some aspects of the manuscript should be revised to improve the quality and increase the novelty of the paper.
Overall comments for improvement:
In introduction, the section of DC (immunostimulatory versus inhibitory molecules) could be improve by adding a figure showing DC-T cell interaction.
Page 4, line 151, please give more detail or define immunnostat.
Page 5, line 2223 please clarify if conventional ex vivo-developed cancer vaccines are unable to fully recapitulate the broad DCs response in vitro or in vivo or both.
Page 10, pages 473-4474, please revise the following sentence and correct it accordingly. “Safety of autologous tumor mRNA-pulsed DC vaccines was demonstrated clinical trials in patients with various types of cancer, such as …”
Line 489, please be more specific when talking about T cells response
Page 15, Section 8. Tumor-derived sEV DC vaccines. In this section the authors nicely described the potential use of Tu-sEV for cancer therapy and how they modulate DCs immunogenicity in vitro and in vivo as well as the impact on T cell response. However, the authors did not refer to whether DC treated with Tus-EVs have the potential to be used for cancers treatment. Therefore, the title is not appropriate and should be changed to “Tumor-derived sEV vaccines”
Page 16, line 774, I believe that the authors meant but not MHC II in state of but MHC II. Please review and correct accordingly.
n/a
Author Response
We appreciate Reviewer #1’s effort in reviewing our manuscript. We find the comments constructive and useful. We revised our manuscript accordingly to improve the quality and novelty.
Below please find our point-to-point response to the reviewers’ comments.
Addressing the comments of Reviewer #1
- In the introduction, the section DC (immunostimulatory versus inhibitory molecules) could be improved by adding a figure showing DC-T cell interaction
Response:
The figure showing DC-T cell interaction has been added in page 3. The diagram illustrates DC activation, maturation, and antigen-loading in the tumor microenvironment and DC migration to tumor-draining lymph nodes to cross-present and cross-prime CD8+ T cells.
- Page 4, line 151, please give more detail or define immunostat.
Response:
The sentence in lines 149-152 is corrected as follows: “The limited efficacy of cancer vaccines may be mainly due to the presence of multiple immunosuppressive factors in the TME that act as immune rheostats or immunnostats in short. In other words, the immunosuppressive factors modulate the antitumor T cell responses and act as a common rate-limiting step in clinical studies that altogether limit the clinical efficacies of cancer vaccines.”
- Page 5, lines 222-223, please clarify if conventional ex vivo-developed cancer vaccines are unable to fully recapitulate the broad DCs response in vitro or in vivo or both.
Response:
The sentence in lines 222-223 is corrected as follows: “Conventional ex vivo-developed cancer vaccines are unable to fully recapitulate the broad DC responses in vivo.”
- Page 10, lines 473-474, please revise the following sentence and correct it accordingly. “Safety of autologous tumor mRNA-pulsed DC vaccines was demonstrated clinical trials in patients with various types of cancer, such as …”
Response:
The sentence in lines 473-474 is corrected as follows: “The safety of autologous tumor mRNA-pulsed DC vaccines was demonstrated in clinical trials of various cancer types, including renal cell carcinoma (Phase I, NCT00006431) and advanced malignant melanoma (Phase I/II, NCT01278940) [235,236].”
- Line 489, please be more specific when talking about T cells response.
Response:
The sentence in lines 486-489 is corrected as follows: “Although the safety profiles of in vitro generated mRNAs have been demonstrated in numerous clinical studies (Phase I/II, NCT00243529; Phase II, NCT00965224, NCT01446731, NCT02285413) [219,239–242], general clinical efficacies have been low due to weak induction of T cell activation [220].”
- Page 15, Section 8. Tumor-derived sEV DC vaccines. In this section the authors nicely described the potential use of Tu-sEV for cancer therapy and how they modulate DCs immunogenicity in vitro and in vivo as well as the impact on T cell response. However, the authors did not refer to whether DC treated with Tu-sEVs have the potential to be used for cancers treatment. Therefore, the title is not appropriate and should be changed to “Tumor-derived sEV vaccines”
Response:
Section 8 discusses the therapeutic effects of tumor-derived sEVs and the modulation of dendritic cells and T cells to achieve such effects, hence its title: “Tumor-derived sEV DC vaccines”. While this section does imply the importance of DC functions in achieving therapeutic effects of Tu-sEVs, I agree that it rather highlights the role of Tu-sEVs as vaccines themselves than their interactions with DCs. The title of section 8 is therefore changed to: “Tumor-derived sEV vaccines”.
- Page 16, line 774, I believe that the authors meant but not MHC II in state of but MHCII. Please review and correct accordingly.
Response:
The sentence in lines 773-775 is corrected to: “Many cancer immunotherapies have focused on inducing MHCI I-restricted tumor-specific CTL responses because most tumor cells constitutively express MHC I, but not MHC II”.
Reviewer 2 Report
The manuscript titled "Dendritic Cell Vaccines: A Shift from Conventional Approach to New Generations" is a well-tried and exhaustive review article by the authors and describes the benefits of many new generation dendritic cell vaccines against the conventional ones. The manuscript further discusses the Advantages and limitations of conventional MoDC-based vaccines and new generations of DC Vaccines and the literature is also inclusive. The manuscript is suitable for publication in the journal although its publication in an immunotherapy-exclusive journal would have been more useful and rewarding.
Author Response
Addressing the comments of Reviewer #2
- The manuscript is suitable for publication in the journal although its publication in an immunotherapy-exclusive journal would have been more useful and rewarding.
Response:
First of all, we appreciate Reviewer #2’s positive comment on the appropriateness of our manuscript to be published in Cells. This review is to provide an update on the new generations of dendritic cell vaccines and perspectives for future cancer therapeutics. We believe that this review could provide new insights to researchers regardless of their research specificity within the broad cancer biology field.
We are currently investigating cancer-immune interactions in the tumor immune microenvironment via small extracellular vesicles, specifically focusing on the interaction-induced changes in dendritic cell functions. This sparked our interest in the clinical implications of our research findings and motivated us to study the most recent research papers that aim to harness dendritic cell functions to develop therapeutic and preventative cancer vaccines.
While preparing this manuscript, we ourselves have also realized the importance of learning new therapeutic strategies to extend our research to clinical implications. We therefore hope that the publication of our manuscript in Cells would allow other researchers, including those who are in a broader cancer research field, to expand the horizon of their research and apply their knowledge to translational work.